# I$^2$C: Intra- and Inter-modality Consistency Learning for Multimodal Sentiment Analysis

## Abstract

Multimodal sentiment analysis (MSA) aims to predict human sentiments by integrating signals from different modalities such as text, video, and audio. However, sentiment cues are often semantically inefficient—exhibiting inconsistency within and across modalities—that hinders robust understanding and inflates computation. In this paper, we propose I$^2$C, a framework that explicitly models Intra- and Inter-modality Consistency to guide effective and efficient sentiment prediction. I$^2$C first projects token-level features into a shared sentiment space and computes intra- and inter-modality consistency scores (I$^2$CS). The I$^2$CS serves three functions: (1) as a consistency loss for regularizing training; (2) as token-wise weights for reweighting features; and (3) as a compression signal for eliminating redundant or conflicting tokens. Extensive experiments are conducted on the CMU-MOSI and CMU-MOSEI datasets, and the results show that I$^2$C outperforms previous state-of-the-art models. Despite removing 90% of tokens, I$^2$C maintains comparable performance, exhibiting remarkable robustness across varying token budgets. All results highlight consistency-aware learning as an effective strategy to improve the accuracy and efficiency of sentiment prediction.

## 1 Introduction

Multimodal Sentiment Analysis (MSA) aims to infer human sentiments by integrating heterogeneous signals such as spoken language, facial expression, and vocal prosody (Geetha et al., 2024; Yang et al., 2022a). With the growing availability of richly annotated datasets and advances in representation learning, recent approaches, especially Transformer-based approaches, have achieved notable progress by leveraging token-level features from multiple modalities (Wang et al., 2025; Wu et al., 2024). These models typically employ transformer-based encoders followed by cross-modal attention mechanisms to capture temporal and semantic dependencies. Despite the architectural sophistication of current MSA models, they often suffer from semantic inconsistency, arising both within and across modalities. Effectively extracting consistent intra- and inter-modality features remains a critical challenge in MSA and multimodal learning.

While recent advances in multimodal learning have leveraged both intra- and inter-modality information, most approaches treat them as independent modeling targets. Intra-modality modeling typically preserves local structure and modality-specific semantics through self-attention, recurrence, or disentangled representations (Tsai et al., 2018; Hazarika et al., 2020; Wang et al., 2025). Inter-modality modeling focuses on aligning cross-modal cues via cross-attention, contrastive learning, or shared embedding spaces (Yang et al., 2022a;b; Li et al., 2023; Yang et al., 2024; Wu et al., 2024). A common paradigm is to decouple modality-invariant and modality-specific components to balance semantic overlap and diversity. However, such methods often overlook the semantic inconsistencies that arise from redundant intra-modal signals or conflicting cross-modal cues, which can introduce representational noise and impair fusion. Without explicitly addressing these inconsistencies, the learned features may remain inconsistent or inefficient under real-world conditions, resulting in poor sentiment prediction performance.

To this end, as shown in Fig. 1, we revisit the disentanglement paradigm in MSA: *if performance is hindered by semantic inconsistency within and across modalities, why not directly optimize intra- and inter-modality consistency during training, rather than relying on indirect disentanglement?* In particular, inspired by prior work that performs token compression based on attention or entropy

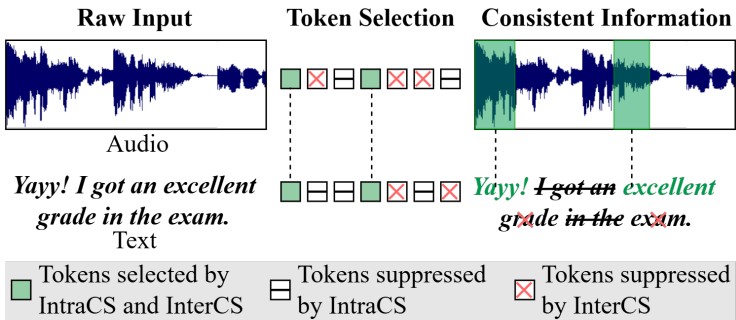

Figure 1: Our insight is to utilize intra-modality consistency (IntraCS) and inter-modality consistency (InterCS) for selecting informative tokens to effectively capture human sentiment.

(Zhao et al., 2025), we are motivated to further explore a consistency-guided compression strategy, leveraging the learned intra- and inter-modality consistency score ($I^2CS$) to enhance both efficiency and accuracy in MSA. Importantly, we aim to measure consistency in the semantic space rather than the token space, where attention is typically applied.

To achieve consistency-aware learning, we propose $I^2C$, a framework that models semantic consistency within and across modalities to guide efficient MSA. $I^2C$ begins by projecting token-level features from each modality into a shared latent sentiment space, where it computes an Intra- and Inter-modality Consistency Score ($I^2CS$) based on the Jensen-Shannon divergence between latent prediction distributions. This consistency score serves as a central signal for three key components: (1) a consistency loss that regularizes model training by enforcing semantic consistency; (2) a token-wise weighting module that highlights informative tokens while reducing the weightage of noisy or ambiguous ones; and (3) a compression mechanism that filters out semantically redundant or conflicting tokens to reduce computational cost. Through this integration, $I^2C$ enables informative token selection grounded in semantic consistency, rather than token-level heuristics. Extensive experiments on CMU-MOSI and CMU-MOSEI demonstrate that $I^2C$ achieves state-of-the-art (SOTA) performance, while maintaining comparable or even better results under significant token sparsity (retaining just 10% of tokens), establishing consistency as a powerful signal for robust and efficient MSA. In summary, our contributions are as follows:

- We define an intra- and inter-modality consistency score ($I^2CS$) to jointly model semantic consistency within and across modalities for MSA. Especially, $I^2CS$ is defined in the sentiment prediction distribution space.
- We propose the $I^2C$ framework, which leverages the $I^2CS$ to jointly guide loss regularization and token-level feature reweighting or compression for MSA tasks. By optimizing both intra- and inter-modality consistency, $I^2C$ enables effective token compression, leading to more efficient and accurate prediction.
- $I^2C$ outperforms the state-of-the-art methods on two datasets. The representations learned using our method demonstrate robustness to significant token compression.

## 2 RELATED WORK

**Disentangled Multimodal Sentiment Analysis.** Recent work in MSA has explored disentangling representations to better separate shared semantics from modality-specific patterns. Tsai et al. (2018) introduce a factorized approach that isolates private and common components, enabling clearer modeling of audio and visual information. MISA (Hazarika et al., 2020) extends this idea by projecting each modality into both shared and unique subspaces, effectively reducing modality disparity and encouraging diverse representations. Building on this, Yang et al. (2022a;b) combine metric learning with adversarial strategies to align invariant and specific features, improving fusion quality. DMD (Li et al., 2023) further addresses distribution mismatch via a decoupled distillation mechanism across modalities. RSA-Net (Gedamu et al., 2023) captures intra- and inter-frame spatial–temporal relations within skeleton sequences using disentangled self-attention, but it remains a

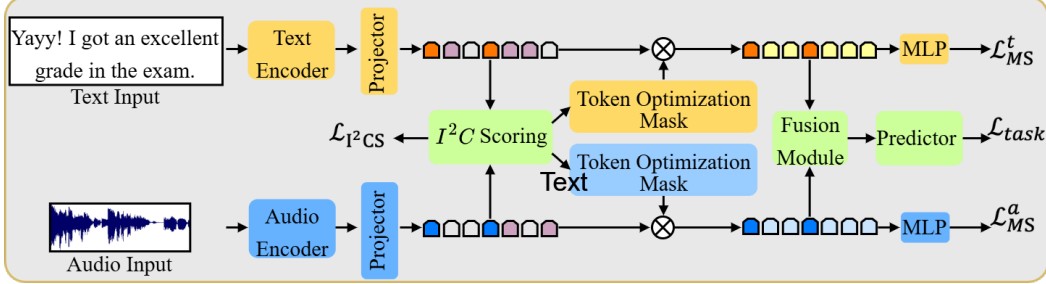

Figure 2: Overview of the I²C framework. Given raw audio and text inputs, modality-specific encoders followed by a projector extract contextual tokens. The I²C Scoring projects these tokens into a sentiment space and computes semantic-level intra- and inter-modality consistency scores, which are used to generate token optimization masks for selective compression. These masks guide both modality-specific heads and multimodal fusion. Meanwhile, the consistency scores also contribute to the consistency-guided loss $\mathcal{L}_{\text{I²CS}}$, enabling the model to learn discriminative yet consistent token representations. The model is trained end-to-end with a joint loss combining task prediction, consistency regularization, and modality-specific supervision.

single-modality relation-mining framework in which the attention mechanism is operated in feature space. The DLF framework (Wang et al., 2025) adopts structured projections with targeted disentanglement objectives to strengthen language-focused representation learning. While these approaches enhance modular representation and cross-modal alignment, they largely overlook the explicit modeling of semantic consistency within and across modalities, especially in the prediction distribution space—an essential factor for robust and efficient MSA.

**Token Compression in Multimodal Learning.** To improve computational efficiency, some studies have explored selecting or compressing informative tokens in multimodal architectures. Early approaches primarily relied on attention weights to identify salient regions or frames within each modality (He et al., 2024; Chen et al., 2024), assuming that high-attention tokens contribute more to the final prediction. Other methods measure token importance via entropy or confidence margins (Xing et al., 2025), aiming to retain only the most certain or informative inputs during training or inference. Recent advances further introduce learnable token selection modules (Tong et al., 2024; Zhang et al., 2024), which dynamically prune less informative tokens based on distributional signals or latent relevance. While effective, these strategies operate mainly in the token or feature space, and often lack explicit modeling of high-level semantic consistency across modalities. In contrast, our work focuses on consistency in the semantic space, offering a unified signal to guide both token weighting and compression based on latent prediction agreement within and across modalities.

## 3 METHOD

### 3.1 PRELIMINARIES

Let $x^{(a)} \in \mathbb{R}^{L_a}$ and $x^{(t)} \in \mathbb{R}^{L_t}$ denote the raw audio waveform and the text sequence, respectively. The task of multimodal sentiment analysis (MSA) is to predict a sentiment label $y$ (either categorical or continuous) given the multimodal input pair $(x^{(a)}, x^{(t)})$.

The raw inputs are encoded into token-level features $h^{(a)} \in \mathbb{R}^{B \times T \times D}$ and $h^{(t)} \in \mathbb{R}^{B \times T \times D}$ (for simplicity, we omit the superscript notation and refer to $h_i$ throughout the rest of this paper.), where $B$ is the batch size, $T$ is the token length, and $D$ is the feature dimension. These embeddings capture local acoustic and linguistic cues and serve as the input to a fusion module that models cross-modal interactions. The goal of MSA is thus to infer the sentiment label $y$ by identifying and aggregating sentiment-relevant signals across modalities.

## 3.2 OVERVIEW

Figure 2 illustrates the overall architecture of the proposed I$^2$C framework, which explicitly models and exploits intra- and inter-modality consistency for efficient multimodal sentiment analysis (MSA). Given raw text and audio inputs, modality-specific encoders first transform them into token-level feature sequences. These representations are then passed to the *I$^2$C Scoring* module, which computes semantic consistency scores across and within modalities. The resulting scores are used to generate *token optimization masks* that softly or sparsely reweight input tokens based on their relevance and consistency.

The masked features are then fed into modality-specific predictors (via lightweight MLPs) to compute unimodal losses, as well as into a fusion module that captures cross-modal interactions for final prediction. The I$^2$C loss, derived from consistency scores, serves as an auxiliary training objective that encourages consistent token representations while suppressing redundancy and conflicts. In this way, I$^2$C simultaneously improves performance and reduces computational cost by selectively retaining informative and semantically aligned tokens.

## 3.3 INTRA- AND INTER-MODALITY CONSISTENCY SCORE (I$^2$CS)

In MSA, redundancy and conflict are two pervasive challenges that hinder robust representation learning. Redundancy arises when semantically similar or neutral tokens dominate a modality, diluting sentiment signals; conflict occurs when different modalities convey contradictory emotional cues, such as a positive utterance with a sarcastic tone.

Instead of separately detecting and mitigating redundancy or conflict, we propose a unified perspective: both phenomena can be interpreted as manifestations of semantic inconsistency—either within a single modality or between modalities. From this viewpoint, learning to maximize *intra- and inter-modality consistency* becomes a more structured and principled solution.

To this end, we introduce a scoring mechanism based on Jensen–Shannon (JS) divergence (Fuglede & Topsoe, 2004) to measure semantic consistency across token-level latent prediction distributions. Given two normalized distributions $p$ and $q$ (e.g., semantic-level latent predictions), the JS divergence is defined as:

$$\text{JS}(p\|q) = \frac{1}{2}\text{KL}(p\|m) + \frac{1}{2}\text{KL}(q\|m), \quad m = \frac{1}{2}(p+q), \tag{1}$$

where the Kullback-Leibler (KL) divergence is computed as:

$$\text{KL}(p(x)\|q(x)) = \sum_x p(x)\log\frac{p(x)}{q(x)}. \tag{2}$$

In our formulation, $p$ and $q \in \mathbb{R}^{B \times L \times C}$ denote normalized prediction distributions, where $B$ is the batch size, $L$ is the token length, and $C$ is the number of semantic categories (e.g., sentiment classes). Compared to KL divergence, JS divergence is symmetric, always finite, and bounded, making it suitable for consistency scoring across modalities.

Based on the JS divergence, we define the I$^2$CS, which consists of two parts, jointly modeling the intra- and inter-modality consistency.

- **Intra-modality Consistency Score** measures intra-modality consistency between current token $h_i$ and other tokens $h_j(j \neq i)$ in the same modality, it is defined as:

$$\text{IntraCS}(h_i) = \frac{1}{T-1}\sum_{j\neq i}\text{JS}(p_i\|p_j), \tag{3}$$

where $p_i, p_j$ are the distribution of latent sentiment prediction like `Softmax()`, $T$ is the token length.

- **Inter-modality Consistency Score** measures the inter-modality consistency between the current token $h_i$ in a source modality $p_i^{\text{src}}$ and the aligned tokens $p_j^{\text{align}}$ in other modalities.

$$\text{InterCS}(h_i) = \frac{1}{N}\sum_{n=1}^{N}\frac{1}{T}\sum_{j=1}^{T}\text{JS}(p_i^{\text{src}}\|p_j^{\text{align}}), \tag{4}$$

where $p_i^{\text{src}}$ is the $i$th token's prediction distribution from the source modality and $p_j^{\text{align}}$ is the corresponding token's distribution in paired modality. $T$ is the token length. $N$ represents the number of modality pairs involved. For a setting with $M$ modalities, $N = M - 1$ (e.g., for 3 modalities, $N = 2$).

Our I$^2$CS is further formulated as:

$$\text{I}^2\text{CS}(h_i) = \alpha \cdot \text{IntraCS}(h_i) + \beta \cdot \text{InterCS}(h_i) \tag{5}$$

where $\alpha$ and $\beta$ are weighting parameters that balance the intra- and inter-modality consistency, respectively.

This consistency score guides representation optimization via loss regularization and token selection, leading to more compact yet expressive multimodal features.

### 3.4 I$^2$CS-based Consistency Loss

To facilitate the learning objective, as shown in Fig. 3, we slightly adjust the I$^2$CS score by penalizing relevant tokens, resulting in I$^2$CS loss as following:

$$\mathcal{L}_{\text{I}^2\text{CS}} = \frac{1}{T} \sum_{i=1}^{T} \text{I}^2\text{CS}(h_i) \cdot \text{Rel}(h_i)$$

$$= \frac{1}{T} \sum_{i=1}^{T} [\alpha \cdot \text{IntraCS}(h_i) + \beta \cdot \text{InterCS}(h_i)] \cdot \text{Rel}(h_i) \tag{6}$$

where $\text{Rel}(h_i)$ measures relevance, which indicates the confidence of the latent feature prediction:

$$\text{Rel}(h_i) = \max(p_i) \tag{7}$$

where $p_i$ is the distribution of latent sentiment prediction like `Softmax()`.

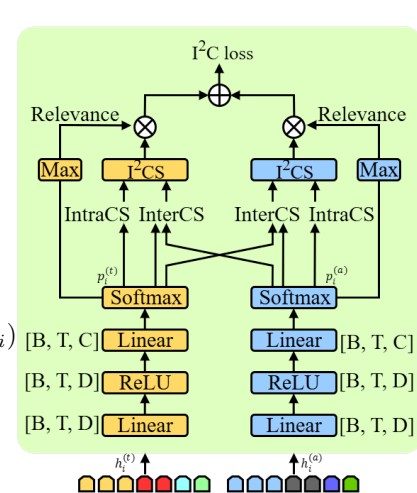

Figure 3: Details of our I$^2$C Scoring module. It projects Text and Audio tokens into a semantic space and computes IntraCS, InterCS, and Relevance for computing $\mathcal{L}_{\text{I}^2\text{CS}}$.

### 3.5 Token Optimization Mask via I$^2$CS

The I$^2$CS values are further utilized to guide token reweighting (soft) and token compression (hard). We first define a semantic importance signal for each token as:

$$\text{Signal}(h_i) = \text{Rel}(h_i) - \text{I}^2\text{CS}(h_i) \tag{8}$$

where tokens with higher semantic confidence and higher consistency (lower I$^2$CS indicates higher consistency) will receive higher signal values, indicating higher importance for sentiment prediction.

**Soft selection.** In the soft setting, we apply a sigmoid gating mechanism to generate continuous importance weights:

$$w_i = \sigma(\gamma \cdot \text{Signal}(h_i)), \quad w_i \in (0, 1) \tag{9}$$

where $\sigma$ is the Sigmoid function, $\gamma$ is a sharpness factor controlling the gate steepness. Thus the soft mask will be $M_{soft} = [w_1, w_2, ..., w_T]$. These weights are used to softly reweight token features for downstream prediction.

**Hard selection.** For hard selection, we apply a *TopK* selection to compress the number of tokens:

$$m_i = TopK[w_i] \tag{10}$$

Only $K$ tokens with the *Top K* scores are retained for subsequent computations. Thus the hard mask will be $M_{hard} = [m_1, m_2, ..., m_K]$. This enables fine-grained token-level compression while preserving semantically critical information.

As shown in Fig. 2, the masks are derived from Eqs. (8)$\sim$(10), after which the optimized tokens are concatenated for the final prediction.

### 3.6 FINAL LEARNING OBJECTIVE

**Task loss.** To supervise the learning, we adopt a standard $\ell_1$ regression loss. Given the predicted sentiment score $\hat{y} \in \mathbb{R}^N$ and the ground-truth label $y \in \mathbb{R}^N$, the task loss is defined as:

$$\mathcal{L}_{\text{task}} = \frac{1}{N} \sum_{i=1}^{N} |\hat{y}_i - y_i| \tag{11}$$

where $N$ is the number of samples. This loss encourages the predicted sentiment to align closely with the human-annotated labels and is commonly used in multimodal sentiment analysis when the target is a continuous value.

**Modality-specific losses (MS losses).** In addition to the task loss and $I^2C$-based loss, we further compute the MS losses before original multimodal feature fusion. The MS losses are formulated as follows:

$$\mathcal{L}_{\text{MS}} = \sum_{m \in \{t,a\}} \lambda_m \cdot l_m = \sum_{m \in \{t,a\}} \lambda_m \cdot |\hat{y}_m - y_{\text{true}}| \tag{12}$$

where $\hat{y}_m$ is the MS prediction, and $y_{\text{true}}$ is the label.

**Joint learning objective.** To sum up, we design the joint loss as the final learning objective:

$$\mathcal{L}_{\text{joint}} = \lambda_1 \mathcal{L}_{\text{task}} + \lambda_2 \mathcal{L}_{I^2\text{CS}} + \lambda_3 \mathcal{L}_{\text{MS}} \tag{13}$$

where $\lambda_n$ is the hyperparameter which controls the relative importance of different losses.

## 4 EXPERIMENTS

### 4.1 DATASETS AND EVALUATION METRICS

We conduct experiments on two widely-used MSA datasets: **CMU-MOSI** (Zadeh et al., 2016) and **CMU-MOSEI** (Zadeh et al., 2018b), both containing annotated video clips with aligned audio, visual, and textual information.

**CMU-MOSI.** This dataset consists of 2,199 short video segments, where each segment features a speaker expressing an opinion in monologue form. The data split includes 1,284 training instances, 229 for validation, and 686 for testing.

**CMU-MOSEI.** As a larger-scale benchmark, MOSEI contains 22,856 video clips collected from online movie reviews. Similar to MOSI, each clip is annotated with fine-grained sentiment labels on a continuous scale from $-3$ (strongly negative) to $+3$ (strongly positive). The dataset adopts a commonly used partitioning: 16,326 samples for training, 1,871 for validation, and 4,659 for testing.

**Evaluation metrics.** Following standard evaluation protocols in prior studies (Liang et al., 2021; Lv et al., 2021; Mao et al., 2022), we assess model performance using several metrics. These include 7-class accuracy (Acc-7), binary accuracy (Acc-2), F1 score, and mean absolute error (MAE). These metrics provide a comprehensive view of the model's effectiveness for sentiment analysis.

### 4.2 IMPLEMENTATION DETAILS

We implement all experiments using PyTorch on a single NVIDIA A100 GPU with 40GB of memory. For the text modality, we adopt RoBERTa (Liu et al., 2019) as the encoder, generating 1024-dimensional word-level embeddings. For the audio modality, we employ Data2Vec-Audio (Baevski et al., 2022), which produces 768-dimensional token representations. The $I^2C$ model is trained using the AdamW optimizer with an initial learning rate of $5 \times 10^{-6}$ and a batch size of 8. Following the training setup in (Wu et al., 2024), we train the model until convergence, equipped with early stopping with a patience of 8 epochs. Additionally, we also utilize BERT (Devlin et al., 2019) as the text encoder to compare with previous BERT-based SOTA models. Please refer to the Appendix for more details.

Table 1: Comparison of our method with prior SOTA methods on MOSI and MOSEI datasets. The best performance is highlighted in bold. T: Text, A: Audio, and V: Video. Note: [†] denotes the result from THUIAR's GitHub page, [*] denotes the result from (Hazarika et al., 2020), - denotes the result from the original paper is not provided, and others are all from the original papers. For fair comparison, the BERT-based models utilize T+A+V following DLF, while the RoBERTa-based model utilizes T+A following MMML.

| Method | Modality | CMU-MOSI | | | | CMU-MOSEI | | | |
|---|---|---|---|---|---|---|---|---|---|
| | | Acc-7($\uparrow$) | Acc-2($\uparrow$) | F1($\uparrow$) | MAE($\downarrow$) | Acc-7($\uparrow$) | Acc-2($\uparrow$) | F1($\uparrow$) | MAE($\downarrow$) |
| BERT-based | | | | | | | | | |
| TFN* | T+A+V | 34.90 | 80.08 | 80.07 | 0.901 | 50.20 | 82.50 | 82.10 | 0.593 |
| LMF* | T+A+V | 33.20 | 82.50 | 82.40 | 0.917 | 48.00 | 82.00 | 82.10 | 0.623 |
| EF-LSTM[†] | T+A+V | 35.39 | 78.48 | 78.51 | 0.949 | 50.01 | 80.79 | 80.67 | 0.601 |
| LF-DNN[†] | T+A+V | 34.52 | 78.63 | 78.63 | 0.955 | 50.83 | 82.74 | 82.52 | 0.580 |
| MFN[†] | T+A+V | 35.83 | 78.87 | 78.90 | 0.927 | 51.34 | 82.85 | 82.85 | 0.575 |
| Graph-MFN[†] | T+A+V | 34.64 | 78.35 | 78.35 | 0.956 | 51.37 | 83.48 | 83.43 | 0.575 |
| MulT | T+A+V | 40.00 | 83.00 | 82.00 | 0.871 | 51.80 | 82.50 | 82.30 | 0.580 |
| PMR | T+A+V | 40.60 | 83.60 | 83.60 | - | 52.50 | 83.60 | 83.40 | - |
| MISA[†] | T+A+V | 41.37 | 83.54 | 83.58 | 0.777 | 52.05 | 84.67 | 84.66 | 0.558 |
| MAG-BERT | T+A+V | 43.62 | 84.43 | 84.61 | 0.727 | 52.67 | 84.82 | 84.71 | 0.543 |
| FDMER | T+A+V | 44.10 | 84.60 | 84.70 | 0.724 | 54.10 | 86.10 | 85.80 | 0.536 |
| DMD | T+A+V | 45.60 | 86.00 | 86.00 | - | 54.50 | 86.60 | 86.60 | - |
| MMIM | T+A+V | 46.65 | 86.06 | 85.98 | 0.700 | 54.24 | 85.97 | 85.94 | 0.526 |
| Self-MM[†] | T+A+V | 46.67 | 85.98 | 85.95 | 0.713 | 53.87 | 85.17 | 85.30 | 0.530 |
| DLF | T+A+V | 47.08 | 85.06 | 85.04 | 0.731 | 53.90 | 85.42 | 85.27 | 0.536 |
| **I$^2$C (Ours)** | T+A+V | **47.96** | **86.06** | **86.06** | **0.698** | **54.92** | **86.67** | **86.65** | **0.525** |
| RoBERTa-based | | | | | | | | | |
| MMML | T+A | 50.34 | 89.69 | 89.67 | 0.580 | 55.74 | 88.02 | 88.15 | 0.492 |
| **I$^2$C (Ours)** | T+A | **51.46** | **90.09** | **90.08** | **0.557** | **56.00** | **88.41** | **88.39** | **0.485** |

## 4.3 RESULTS

**Baselines.** To assess the effectiveness of our proposed approach, we compare I$^2$C against a comprehensive set of competitive baselines on both the CMU-MOSI and CMU-MOSEI datasets. The selected methods span a broad spectrum of representative MSA strategies, including early fusion (**EF-LSTM**(Williams et al., 2018b), **LF-DNN**(Williams et al., 2018a)), memory-augmented frameworks (**MFN**(Zadeh et al., 2018a), **Graph-MFN**(Zadeh et al., 2018b)), attention-based and transformer-based approaches (**MulT**(Tsai et al., 2019), **TFN**(Zadeh et al., 2017), **LMF**(Liu et al., 2018), **MAG-BERT**(Rahman et al., 2020), and **PMR**(Lv et al., 2021)), multi-task learning (**Self-MM** (Yu et al., 2021)), as well as more recent decoupled or multi-loss techniques (**MISA**(Hazarika et al., 2020), **FDMER**(Yang et al., 2022a), **DMD**(Li et al., 2023), **DLF**(Wang et al., 2025), **MMIM**(Han et al., 2021), and **MMML**(Wu et al., 2024)). Please refer to the Appendix for more details.

**Performance Comparison.** Table 1 shows that our I$^2$C framework consistently outperforms a wide spectrum of prior MSA models on both CMU-MOSI and CMU-MOSEI. Compared with earlier fusion-based approaches, I$^2$C achieves substantial improvements, and even when set against the strongest recent baselines, it establishes new state-of-the-art performance across all evaluation metrics. Importantly, the gains are not confined to a specific backbone: compared with both previous BERT-based SOTA models and RoBERTa-based MMML, which is the SOTA model, I$^2$C demonstrates stable and significant advances.

**Discussion.** These results yield several insights. *First*, semantic consistency-guided learning is effective: by suppressing redundancy and resolving inter-modal conflicts, I$^2$C strengthens both fine-grained sentiment classification and regression stability. *Second*, the framework is complementary to stronger unimodal encoders—its benefits persist even when paired with RoBERTa, indicating

Table 2: Results of ablation studies on the MOSI benchmark.

| Method | Acc-7($\uparrow$) | Acc-2($\uparrow$) | F1($\uparrow$) | MAE($\downarrow$) |
|---|---|---|---|---|
| $I^2C$ (Ours) | **51.46** | **90.09** | **90.08** | **0.557** |
| w/o IntraCS | 47.52 | 87.20 | 87.21 | 0.629 |
| w/o InterCS | 46.79 | 88.87 | 88.83 | 0.626 |
| w/o Rel. in Equ.(6) | 45.48 | 87.96 | 87.99 | 0.668 |
| w/o Rel. in Equ.(8) | 46.36 | 87.80 | 87.82 | 0.647 |
| w/o Gating | 45.77 | 87.20 | 87.21 | 0.653 |
| w/o $\mathcal{L}_{MS}$ | 47.38 | 87.50 | 87.51 | 0.643 |
| w/o $\mathcal{L}_{I^2CS}$ | 48.83 | 86.59 | 86.62 | 0.634 |

Table 3: Further analysis on MOSI.

(a) Different token retention ratios.

| Ratio | Acc-7($\uparrow$) | Acc-2($\uparrow$) | F1($\uparrow$) | MAE($\downarrow$) |
|---|---|---|---|---|
| 1.0 | 51.46 | 90.09 | 90.08 | 0.557 |
| 0.8 | **52.92** | **90.85** | **90.84** | **0.553** |
| 0.3 | 52.04 | 89.33 | 89.31 | 0.567 |
| 0.1 | 49.42 | 89.18 | 89.17 | 0.576 |

(b) Different masking signals.

| Method | Acc-7($\uparrow$) | Acc-2($\uparrow$) | F1($\uparrow$) | MAE($\downarrow$) |
|---|---|---|---|---|
| $I^2C$ (Ours) | **51.46** | **90.09** | **90.08** | **0.557** |
| Entropy | 50.29 | 88.72 | 88.75 | 0.578 |
| Attention | 50.48 | 88.63 | 88.62 | 0.572 |
| Random | 49.27 | 88.33 | 88.31 | 0.575 |

generality rather than dependence on a weaker backbone. *Third*, the fact that a text+audio configuration of $I^2C$ with stronger encoders can outperform many text+audio+video baselines and the SOTA text+audio MMML underscores its efficiency and robustness, showing that the model can leverage informative modalities while mitigating noisy ones. Overall, the findings highlight $I^2C$ as a versatile, generalizable, and efficient approach for MSA.

## 4.4 ABLATION STUDY

We conduct ablation studies on the MOSI dataset to evaluate the effectiveness of different critical components in our $I^2C$ framework. Specifically, we analyze the contributions of intra- and inter-modality consistency scores (IntraCS and InterCS), relevance estimation (Rel. in Equ.(6) and (8)), reweighting mechanisms (gating), and both modality-specific and consistency-guided training objectives ($\mathcal{L}_{MS}$ and $\mathcal{L}_{I^2CS}$).

**Different components.** To understand the contribution of each component in our $I^2C$ framework, we conduct component-wise ablations on the MOSI dataset, as summarized in Table 2. Removing either intra- or inter-modality consistency (IntraCS, InterCS) significantly reduces performance across all metrics, highlighting the necessity of modeling consistency across and within modalities. Furthermore, excluding the relevance terms (Eq. (6) and Eq. (8)) also degrades performance, indicating that modeling token-level sentiment relevance is critical for consistency-aware learning. Lastly, removing gating mechanism in Equ. (9) or either of the learning losses ($\mathcal{L}_{MS}$ or $\mathcal{L}_{I^2CS}$) consistently leads to performance drops, validating that our training objective and adaptive token optimization strategy synergize well to improve robustness and effectiveness.

## 4.5 FURTHER ANALYSIS

**Token Compression Robustness Analysis.** We evaluate the robustness of our model under different token retention ratios ranging from 1.0 to 0.1. As shown in Table 3a, the overall performance remains stable even under substantial token reduction. Interestingly, a moderate retention ratio (e.g., 0.8) even outperforms the 1.0 setting across several metrics. This aligns with our hypothesis that a small portion of tokens contain redundant or conflicting evidence, and removing them can sharpen the shared sentiment-relevant signal captured by the $I^2C$ scoring mechanism. Even at aggressive compression levels (e.g., 0.3 or 0.1), the model maintains competitive performance. This is because (1) the highest-scoring tokens identified by $I^2C$ tend to carry the core semantic cues, and (2) the

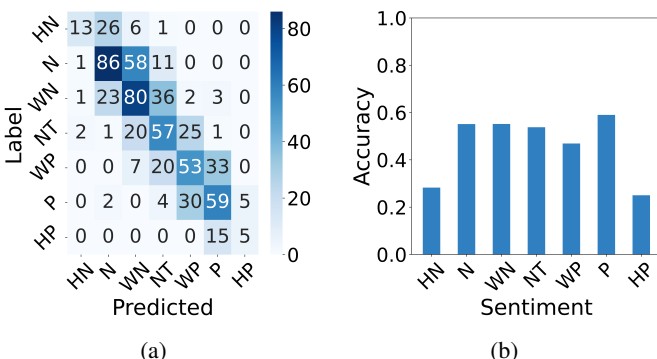

(a)                                                    (b)

Figure 4: Analysis on the MOSI dataset. (a) Confusion matrix, and (b) Corresponding accuracy for each sentiment. HN: Highly Negative; N: Negative; WN: Weakly Negative; WP: Weak Positive; P: Positive; HP: Highly Positive.

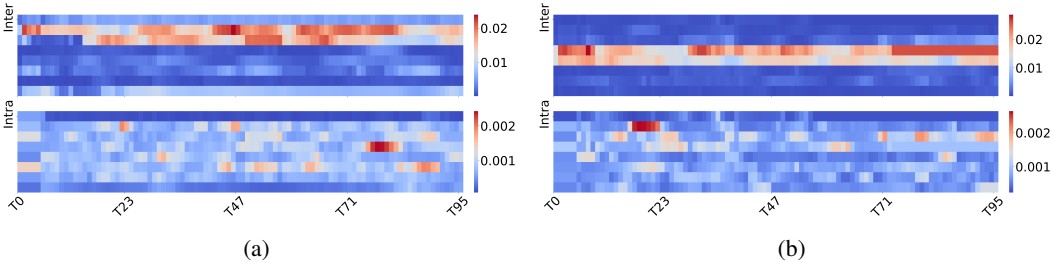

(a)                                                    (b)

Figure 5: Visualization of intra- and inter-modality semantic consistency at the token level. (a) Pre-optimization with $I^2C$ framework. (b) After-optimization with $I^2C$ framework. *Note*: Lower value, higher consistency.

CLS token is not used as a special representation, so discarding a large fraction of tokens does not disrupt modality summarization or fusion. Overall, these results suggest that our method enables robust and semantically informed token compression, allowing the model to remain accurate under constrained computational budgets while effectively filtering inconsistent or low-relevance tokens, demonstrating strong efficiency and scalability under constrained settings.

**Masking Signals.** To further verify the effectiveness of our proposed $I^2CS$-based masking strategies, as formulated in equations (8)∼(10), we compare our $I^2CS$-based masking with three standard heuristics—Entropy, Attention, and Random masking—where tokens with high entropy, low attention weights, or randomly selected tokens are removed, respectively. As shown in Table 3b, these heuristics perform similarly since they do not model semantic consistency. Our $I^2CS$ score considers both intra- and inter-modal consistency, yielding a more reliable signal for identifying redundant or conflicting tokens and therefore achieves the best performance.

**Confusion Matrix.** To further investigate the effect of sentiment granularity on MSA, as shown in Fig.,4, we examine the confusion matrix and per-class accuracy on MOSI. Most sentiment categories achieve accuracies above 50%; however, "HN" and particularly "HP" exhibit substantially lower performance, thereby constraining the overall 7-class accuracy. A closer inspection of the confusion matrix shows that "HN" and "HP" contain only 46 and 20 samples, respectively, underscoring the influence of the dataset's long-tailed distribution. This imbalance poses a critical challenge for fine-grained sentiment understanding and motivates future exploration in this direction.

**Visualizing Token Consistency.** To better assess the effectiveness of $I^2C$, we visualize token-level intra- and inter-modality semantic consistency in Fig. 5. Before optimization, tokens exhibit weak consistency both within and across modalities. After applying our consistency-aware training, consistency improves substantially in both aspects, explaining the robustness of $I^2C$ under aggressive token compression. These results confirm that $I^2C$ enhances representational quality by explicitly modeling and supervising semantic consistency during training.

## 5 CONCLUSION

We propose I$^2$C, a consistency-aware framework for MSA that explicitly models and optimizes intra- and inter-modality token-level semantic consistency. Unlike indirect disentanglement, I$^2$C directly models consistency across and within modalities to guide token optimization by reweighting or compression. Extensive experiments on two popular MSA benchmarks demonstrate the superior performance and robustness of I$^2$C under token budget constraints.

## BROADER IMPACT AND FUTURE WORK

**Broader Impact.** (i) This study reveals the importance of modeling intra- and inter-modality consistency, offering a new perspective for exploiting multimodal learning. (ii) The proposed consistency-aware learning paradigm is general and flexible, allowing I$^2$C to be readily extended to other multimodal tasks by adapting its scoring inputs.
**Future Work.** For missing or degraded modalities, the current consistency modeling and token optimization may not fully apply. Future work will explore robust consistency learning under modality missing settings.

## ETHICAL STATEMENT

Multimodal Sentiment Analysis (MSA) enhances human-computer interaction by enabling machines to understand human emotions across language, vision, and audio. While this technology offers clear benefits, it also raises concerns regarding user privacy, potential algorithmic biases, and the societal impact of rapid deployment. In this work, we utilize two publicly available and widely adopted academic datasets—CMU-MOSI and CMU-MOSEI. All data usage complies with ethical and legal standards, and MOSEI remains the largest benchmark dataset for MSA research.

## REPRODUCIBILITY STATEMENT

To ensure the reproducibility of our work, we provide comprehensive experimental details in the paper and supplementary materials. Specifically, we report the computing infrastructure used for all experiments, including hardware specifications and software environments. We also include the complete set of hyperparameters for each model, training schedule, and evaluation protocol. Furthermore, we summarize the full algorithmic pipeline in the appendix underlying our approach, offering a clear description of each step to facilitate independent verification. All these details enable faithful reproduction of our results and provide a foundation for future research building upon our method.

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

# A  APPENDIX

## A.1  CONFIGURATION OF HYPERPARAMETERS

Table 4 presents the configuration of all hyperparameters of the proposed I$^2$C for each benchmark. These hyperparameters are optimized based on performance evaluations conducted on the validation dataset. It takes from 30 minutes to several hours to train models on one NVIDIA A100-40G GPU.

Table 4: Hyperparameter settings on both MSA benchmarks.

| Setting | MOSI | MOSEI |
|---|---|---|
| Optimizer | AdamW | AdamW |
| Learning rate | 5e-6 | 5e-6 |
| Batch size | 8 | 8 |
| Feature dimension [Text, Audio] | [1024, 678] | [1024,678] |
| Token length | 96 | 96 |
| Early stopping epochs | 8 | 8 |
| $\alpha$ | 1.0 | 1.0 |
| $\beta$ | 1.0 | 1.0 |
| $\gamma$ | 20.0 | 20.0 |
| $\lambda_a$ | 1.0 | 1.0 |
| $\lambda_t$ | 1.0 | 1.0 |
| $[\lambda_1, \lambda_2, \lambda_3]$ | [1, 1, 1] | [1, 0.3, 1] |

## A.2  ALGORITHM AND OVERALL FORMULATION

We summarize the full I$^2$C pipeline in Algorithm 1.

---

**Algorithm 1 End-to-end I$^2$C Pipeline**

---

Input: Raw inputs $x^{(a)}, x^{(t)}$, sentiment label $y$
Output: Prediction $\hat{y}$ and total loss $\mathcal{L}_{\text{joint}}$

---

**Step 1: Feature Encoding**

$$x^{(a)}, x^{(t)} \xrightarrow{\text{Encoders}} h^{(a)}, h^{(t)} \in \mathbb{R}^{B \times T \times D}$$

**Step 2: Semantic Projection**

$$h^{(a)}, h^{(t)} \xrightarrow{\text{Semantic Projector}} p^{(a)}, p^{(t)} \in \mathbb{R}^{B \times T \times C}$$

**Step 3: I$^2$CS Scoring**

$$p^{(a)}, p^{(t)} \xrightarrow{\text{I}^2\text{CS Scorer}} \text{I}^2\text{CS}^{(a)}, \text{I}^2\text{CS}^{(t)} \in \mathbb{R}^{B \times T}$$

**Step 4: Token Masking**

$$\text{I}^2\text{CS}^{(a)}, \text{I}^2\text{CS}^{(t)} \xrightarrow{\text{Masking}} \begin{cases} \tilde{h}^{(a)}, \tilde{h}^{(t)} \in \mathbb{R}^{B \times T \times D}, & \text{(soft)} \\ \tilde{h}^{(a)}, \tilde{h}^{(t)} \in \mathbb{R}^{B \times K \times D}, & \text{(hard)} \end{cases}$$

**Step 5: Prediction and Modality Heads**

$$\tilde{h}^{(a)}, \tilde{h}^{(t)} \xrightarrow{\text{Fusion}} \hat{y}$$

$$\tilde{h}^{(a)} \xrightarrow{\text{MLP}_a} \hat{y}_a, \quad \tilde{h}^{(t)} \xrightarrow{\text{MLP}_t} \hat{y}_t$$

**Step 6: Train using Joint Loss**

$$\mathcal{L}_{\text{joint}} = \lambda_1 \mathcal{L}_{\text{task}} + \lambda_2 \mathcal{L}_{\text{I}^2\text{CS}} + \lambda_3 \mathcal{L}_{\text{MS}}$$

---

Table 5: Performance of different modalities on the MOSI benchmark.

| Method | Acc-7 (%) | Acc-2 (%) | F1 (%) | MAE($\downarrow$) |
|---|---|---|---|---|
| **$I^2$C (Ours)** | **51.46** | **90.09** | **90.08** | **0.557** |
| only Audio | 39.11 | 84.23 | 84.25 | 0.805 |
| only Text | 45.75 | 86.61 | 86.62 | 0.654 |

## A.3 ADDITIONAL RESULTS

**Different modalities.** To evaluate the contribution of each modality in our Text-Audio-based $I^2$C, we conduct ablation experiments by removing either the text or audio modality. As shown in Table 5, removing either modality leads to notable performance degradation. Using only audio results in a sharp drop in Acc-7 and a large MAE increase, indicating its relatively weak standalone predictive power. While using only text yields better performance than audio, it still falls short of the full model. These results underscore the importance of multimodal integration and validate that our consistency-driven fusion effectively leverages complementary information across modalities.

## A.4 THE DETAILS FOR THE THREE MODALITIES IMPLEMENTATION

As shown in Table 1, to ensure fairness, when extending our approach to the three-modality case, we compute the proposed score for each modality by considering its interaction with the other two modalities. Specifically, for a given target modality, we first calculate the score between the target modality and each of the remaining two modalities independently. The final score for this modality is then obtained by averaging these pairwise scores. This averaged score is subsequently used to guide the token optimization process of the target modality. In this way, our method can be naturally generalized from the bimodal setting to the trimodal setting, while maintaining consistent optimization principles.

## A.5 BRIEF INTRODUCTION OF ALL BASELINES

We briefly introduce all baseline methods discussed in the main paper:

- **EF-LSTM**: An early fusion model that concatenates multimodal features and feeds them into a unified LSTM for sentiment prediction.
- **LF-DNN**: This model employs an intermediate fusion approach, combining modality-specific weights during the learning phase.
- **TFN**: It captures intra-modal patterns using three dedicated subnetworks and models inter-modal interactions through unimodal, bimodal, and trimodal subtensors derived via a tensor fusion strategy.
- **LMF**: Utilizes a low-rank tensor approximation to efficiently fuse high-dimensional multimodal data, achieving scalability with the number of modalities.
- **MFN**: Proposes a memory fusion mechanism that performs both view-specific and cross-view processing over sequential inputs, generating predictions from the combined output.
- **Graph-MFN**: Extends MFN by introducing a Dynamic Fusion Graph to model cross-modal interactions as a graph structure, replacing the original Delta-memory Attention Network.
- **MulT**: Applies Transformer layers for soft cross-modal alignment, extending temporal receptive fields and introducing a ternary-symmetric architecture for feature fusion.
- **PMR**: Incorporates a progressive reinforcement process via a message hub and cyclic cross-modal attention, allowing gradual feature refinement among modalities.
- **MISA**: Inspired by domain separation frameworks, it decomposes each modality into modality-shared and modality-private spaces, facilitating structured fusion.
- **MAG-BERT**: Introduces a Multimodal Adaptation Gate that enables BERT to absorb non-verbal cues during the fine-tuning process.

- **FDMER**: Proposes FDMER framework, which disentangles features into modality-invariant (common) and modality-specific (private) subspaces via dedicated encoders and adversarial training, encouraging both shared understanding and modality uniqueness.

- **DMD**: Implements a learnable Graph Unit to dynamically guide knowledge distillation between modalities, enabling adaptable and resilient representation learning.

- **MMIM**: Proposes MultiModal InfoMax (MMIM), a novel framework for multimodal sentiment analysis (MSA) that explicitly maximizes mutual information to preserve task-relevant information during fusion.

- **Self-MM**: Replaces hand-labeled unimodal annotations using a self-supervised label generation module. It jointly trains multimodal and unimodal tasks to learn both consistency and difference across modalities, using dynamic weighting to focus on challenging samples.

- **DLF**: A Disentangled-Language-Focused framework for MSA that explicitly separates shared and modality-specific features, enhances language representations via a language-guided attractor, and adopts hierarchical prediction to improve accuracy.

- **MMML**: Proposes an optimized MSA framework, which integrates carefully selected feature encoders and fusion strategies across modalities. It further introduces multi-loss training and context integration, which significantly enhance representation learning.

## A.6 THE USE OF LARGE LANGUAGE MODELS

In this paper, we utilize large language models (ChatGPT-5) to refine the writing.

