# OpenReview forum: "I$^2$C: Intra- and Inter-modality Consistency Learning for Multimodal Sentiment Analysis"
_ICLR.cc/2026/Conference — Submitted to ICLR 2026_

### Official Review · Reviewer_geLK · 2025-10-16

**Soundness:** 3
**Presentation:** 2
**Contribution:** 3
**Rating:** 2
**Confidence:** 5

**Summary:**

This paper proposes a method to address both intra-modal and inter-modal semantic conflicts.

**Strengths:**

The experimental results are competitive.

**Weaknesses:**

1. The code is not released, and the paper lacks sufficient detail.
2. The novelty is limited. Intra-modal consistency is enforced after the encoder outputs, where token interactions have already occurred via the attention mechanism, making the effectiveness of masking questionable.
3. The inter-modal consistency formulation (Equation 4) is unclear.
4. It is not specified whether the [CLS] tokens from the text and speech encoders are used in subsequent stages—this is crucial for assessing the soundness of the method design.
5. The mathematical notation is confusing, and the experimental explanations are insufficient.
6. In Table 3(a), reducing the parameter from 1 to 0.1 results in only 0.9% performance drop, further questioning the usefulness of post-encoding masking and whether the CLS token is removed.
7. The comparisons in Table 3(b) are unclear.
8. Overall, the paper appears unfinished.

**Questions:**

Refer to the weaknesses listed above.

---

> ### Author Response · Authors · 2025-11-14
> **Response to comments from Reviewer geLK (1/2)**
>
> We appreciate the recognition that the experimental results are competitive. Below, we address each of your questions and concerns in detail.
>
> **[Q1] Code not released; insufficient detail.**
>
> [A1] Thank you. We will release the full codebase, training scripts, and checkpoints upon acceptance (pending legal approval). As for details, the appendix includes hyperparameters and algorithm descriptions, and Reviewer iZXh noted that “the paper is well-structured and logically compelling. It provides a clear explanation of the algorithms and offers a comprehensive set of experiments that thoroughly validate the proposed method,” indicating clarity of presentation.
>
> **[Q2] Novelty concern; effectiveness of masking after a self-attention-based encoder?**
>
> [A2] We appreciate the reviewer's thoughtful comment regarding the novelty. While we agree that self-attention integrates contextual information into token representations, masking in our design remains highly effective and necessary for the following reasons:
>
> - Contextualization vs. Consistency: The encoder's self-attention focuses on feature contextualization (aggregating information). However, it does not guarantee semantic consistency regarding the final sentiment. Even after encoding, individual tokens can still exhibit high prediction uncertainty or conflict with other modalities. Our method specifically identifies and filters these "sentiment-conflicting" tokens in the prediction distribution space.
>
> - Filtering Before Fusion: The goal of our masking is to prevent noisy or conflicting semantics from propagating into the cross-modal fusion stage. Even if representations are contextualized, removing tokens with inconsistent prediction distributions effectively blocks the transmission of ambiguous sentiment signals to the fusion module.
>
> - Empirical Evidence: This design is validated by the ablations in Table 2, which show that removing this post-encoder consistency masking leads to a significant performance drop. In Table 3(b), we also compare our method against the attention-based masking method; the results show that our consistency learning outperforms the attention-based method.
>
> This confirms that despite prior token interactions in the encoder, explicitly enforcing consistency at the sentiment prediction distribution space is crucial and novel for MSA tasks.
>
> **[Q3] Clarification in Equation (4).**
>
> [A3] We updated Eq. (4) as follows:
>
> $$\text{InterCS}(h_i) = \frac{1}{N} \sum_{n=1}^{N} \frac{1}{T} \sum^T_{j=1} \text{JS}(p_i^{\text{src}}|| p_{j}^{\text{align}}),$$ where:
>
> - $p_i^{\text{src}}$ and $p_{j}^{\text{align}}$ denote the sentiment prediction distributions of the $i$-th source token and the $j$-th aligned token, respectively.
>
> - $\text{JS}(\cdot || \cdot)$ is the Jensen-Shannon divergence, which measures the semantic discrepancy between the two distributions.
>
> - $T$ is the token length.
>
> - $N$ represents the number of modality pairs involved. For a setting with $M$ modalities, $N = M - 1$ (e.g., for 3 modalities, $N=2$).
>
> By minimizing this score, we enforce strict semantic alignment for each modality pair in the prediction space.
>
>
> **[Q4] Whether CLS token is used.**
>
> [A4] Thanks. We clarify that our method does not rely on the text encoder’s CLS token at any stage, and therefore the text and speech encoders remain fully comparable even though the speech encoder does not contain a CLS token.
>
> **1. Text encoder (BERT/RoBERTa): CLS is *not* used in our pipeline**
>
> Although BERT outputs a CLS token, in our framework:
>
> - The CLS token is *not* extracted or treated as a sentence embedding.
> - The final text representation is obtained over all remaining tokens, *not* via CLS.
>
> Thus, the existence of a CLS token introduces no special role, no shortcut, and no modality asymmetry.
>
> **2. Speech encoder (Data2Vec-Audio): Naturally no CLS, same token-level processing**
>
> - The speech encoder outputs time-step embeddings only.
> - We apply the same I²CS scoring, gating, and aggregation as in the text modality.
>
> All the above ensure all tokens are comparable at the level where the consistency score operates, guaranteeing the effectiveness of our method.

---

> ### Author Response · Authors · 2025-11-17
> **Response to comments from Reviewer geLK (2/2)**
>
> **[Q5] The mathematical notation is confusing, and the experimental explanations are insufficient.**
>
> [A5] Thanks for the comments. We have revised Equation (4) as Q3 shows, all the revision is highlighted in blue in the revised pdf. We have substantially clarified the experimental section in the revision. In particular, we refined the descriptions in the Further Analysis subsection, where we now:
>
> (1) clearly explain the role of each ablation (token retention, masking signals, and component removal),
>
> (2) explicitly define all baseline masking strategies (entropy, attention, random),
>
> (3) provide detailed interpretations of Table 3(a) and 3(b), and
>
> (4) clarify how token consistency affects performance under different settings.
>
> **[Q6] Why does 0.8 retention outperform 1.0?**
>
> [A6] Thanks. The 1.0 setting passes all tokens after soft gating, whereas the 0.8 setting removes low-quality tokens entirely. This hard removal eliminates residual noise that soft gates cannot fully suppress, explaining why 0.8 slightly outperforms 1.0. In summary, consistency learning provides a soft, continuous regularization that identifies low-quality tokens, while hard removal eliminates them entirely. The small performance gain at a 0.8 retention ratio therefore reflects the complementarity of soft consistency learning and discrete hard pruning. This reinforces, rather than contradicts, the effectiveness of our consistency learning method. Therefore, it should be noted that the performance improvement is from our consistency learning and is not related to the CLS token, which is not used in our operation space.
>
> **[Q7] Table 3(b) comparisons unclear.**
>
> [A7] Thanks for the detailed comments. Table 3(b) examines only the masking signal while keeping all other components constant. We now explicitly clarify the baseline definitions:
>
> - (1) Entropy-based drops tokens with high entropy distributions;
>
> - (2) Attention-based drops tokens with low average attention weights;
>
> - (3) Random drops tokens uniformly at random.
>
> These methods do not model semantic consistency and therefore achieve similar performance. In contrast, our I²C score jointly captures intra- and inter-modal consistency, yielding a more meaningful signal for identifying redundant or conflicting tokens. This is why I²C systematically performs best across all metrics in Table 3(b). We have included this clarification in the revised paper.
>
> **[Q8] Overall, the paper appears unfinished.**
>
> [A8] We apologize if the initial presentation caused any confusion regarding the completeness of our work. We have taken your constructive feedback seriously and have performed a comprehensive revision to ensure the paper is rigorous, clear, and fully finished. Specifically, addressing the concerns raised in Q1–Q7, we have:
>
> - Rigorous Mathematical Formulation: As detailed in [A3] and [A5], we have formally redefined Equation (4) (highlighted in blue in the revision) to explicitly formulate the Inter-modality Consistency Score using Jensen-Shannon divergence, clearly defining the prediction distributions ($p_i^{\text{src}}, p_j^{\text{align}}$) and pairwise interactions ($N$).
>
> - Clarified Model Architecture: We have resolved ambiguities regarding the model structure, specifically clarifying in [A4] and [A2] that our method operates post-encoder without relying on CLS tokens, ensuring full comparability between text and speech modalities.
>
> - Further Experimental Analysis: We have substantially expanded the "Further Analysis" section. This includes:
>
> (1) Explicitly defining baseline strategies (Entropy, Attention, Random) for Table 3(b) as discussed in [A7].
>
> (2) Providing a theoretical interpretation of the retention ratio (0.8 vs. 1.0) in [A6], explaining the complementarity between soft consistency learning and hard pruning.
>
> - Enhanced Reproducibility: As noted in [A1], we have ensured the Appendix contains full hyperparameter details and algorithm descriptions, and we are committed to releasing the full codebase and checkpoints.
>
> With these extensive revisions and clarifications, we believe the manuscript now presents a complete, logically compelling, and mathematically rigorous study.
>
> In conclusion, thanks again for your insightful and detailed comments. We hope our feedback will align with your expectations. It will be very appreciated if you would raise your rating.

---

> > ### Author Response · Authors · 2025-11-20
> > **Further Response to comments from Reviewer geLK**
> >
> > We sincerely appreciate your thoughtful feedback and the time devoted to our submission.
> >
> > We would like to concisely clarify the core innovation of our work. While redundancy and cross-modal conflicts have long been recognized as central challenges in multimodal sentiment analysis, existing methods address them only implicitly—through fusion heuristics, or decoupled architectural design—but none provides an explicit modeling mechanism for semantic consistency itself.
> >
> > Our work introduces a new perspective: we formulate consistency directly at the semantic distribution level, defining both intra- and inter-modality consistency through the proposed I²CS score, a token-level, distribution-based measure that is mathematically grounded and operational in practice. This explicit formulation allows the model to identify redundant or conflicting tokens in a principled way and to leverage consistency as a unified signal for both improved accuracy and efficient token selection. To the best of our knowledge, this is the first explicit, generalizable, and distribution-level consistency modeling framework in MSA.
> >
> > If the clarifications and additional analyses in the rebuttal satisfactorily address the raised concerns, we would be grateful if you could increase the rating. We are happy to provide any further clarification if needed.

---

> ### Author Response · Authors · 2025-11-28
>
> Dear reviewer geLK,
>
> We hope this message finds you well. We have carefully responded to all of the comments and clarifications requested in the reviews. If there are any remaining concerns, we would greatly appreciate it if you could kindly let us know so that we can further clarify. If our responses have addressed your questions, we would be very grateful if you could consider updating your scores accordingly.
>
> Thank you again for your time and feedback.

---

### Official Review · Reviewer_UzNZ · 2025-10-31

**Soundness:** 3
**Presentation:** 3
**Contribution:** 3
**Rating:** 4
**Confidence:** 3

**Summary:**

This paper focuses on Multimodal Sentiment Analysis (MSA), aiming to address the semantic inconsistency problems that exist within and across modalities. Specifically, the authors propose a framework which first projects token-level features into a shared sentiment space and computes intra- and inter-modality consistency scores. The score is calculated based on the Jensen-Shannon (JS) divergence between latent sentiment prediction. Experiments are conducted on the CMU-MOSI and CMU-MOSEI datasets, and the results show that the proposed framework achieves SOTA performance.

**Strengths:**

1. The method achieves state-of-the-art (SOTA) performance on both the CMU-MOSI and CMU-MOSEI benchmark.
2. The paper strongly supports the rationale of the model design through comprehensive ablation studies.

**Weaknesses:**

1. The paper's core motivation hinges on the assertion that "existing methods often overlook the semantic in consistencies that arise from redundant intra-modal signals or conflicting cross-modal cues, which can introduce representational noise and impair fusion". However, this key claim lacks direct theoretical or experimental support.
2. The authors justify their choice of JS divergence by highlighting its advantages over KL divergence. However, the paper lacks a broader justification for why JS divergence is superior to other strong alternatives.  Could the authors elaborate on why this method was chosen over other metrics (e.g., Euclidean distance or Cosine Similarity)?  Have comparative experiments been conducted?"
3. The definition of the Inter-modality Consistency Score is ambiguous. The authors do not clearly explain in the paper how the relevant content in Equation 4 is obtained.
4. The paper lacks a deep analysis of a key finding in Table 3a, where model performance at a 0.8 token retention ratio is superior to the baseline using all tokens (1.0 ratio). This result strongly implies that the 1.0 model is negatively impacted by noisy (redundant or conflicting) tokens. The authors briefly mention this but fail to analyze why the model's soft selection mechanism or consistency loss was not sufficient to automatically suppress this noise.

**Questions:**

Please refer to the weakness

---

> ### Author Response · Authors · 2025-11-14
> **Response to comments from Reviewer UzNZ**
>
> We greatly appreciate your recognition of our state-of-the-art performance, as well as your acknowledgement of the comprehensive ablation studies supporting our model design. Your feedback is encouraging and reinforces the value of our contributions. Below, we respond to each of your concerns in detail.
>
> **[Q1] "existing methods often overlook the semantic inconsistencies that arise from redundant intra-modal signals or conflicting cross-modal cues, which can introduce representational noise and impair fusion.” This claim lacks support.**
>
> [A1] We thank the reviewer for the comment. The claim is supported by both prior studies and our own evidence. Prior work has repeatedly shown that visual and audio streams contain substantial non-semantic noise and that cross-modal conflicts (e.g., sarcasm, misalignment) impair fusion [1-4]. Our ablations further validate this: removing the lowest 20% consistency-scoring tokens improves accuracy (Table 3a in paper), removing the consistency loss reduces performance (Table 2 in paper), and A+V performs worse than T+A/T+V (cross-modal inconsistency) in the following Table. These results empirically confirm that semantic inconsistency is both real and influential.
>
> **Ablation of I²CS pair on the MOSI dataset in the setting of T+A+V.**
>
> | Modality pair | Acc-7 | Acc-2 | F1    | MAE   |
> |-----------|-------|-------|-------|-------|
> | (A, V)    | 43.44 | 83.58 | 83.02 | 0.771 |
> | (V, T)    | 45.92 | 84.49 | 84.50 | 0.732 |
> | (T, A)    | 47.23 | 85.26 | 85.21 | 0.728 |
> | **All (Ours)** | **47.96** | **86.06** | **86.06** | **0.698** |
>
> [1] Misa: Modality-invariant and -specific representations for multimodal sentiment analysis. ACM MM, 2020.
>
> [2] Multimodal transformer for unaligned multimodal language sequences. ACL 2019
>
> [3] Multimodal multi-loss fusion network for sentiment analysis. NAACL 2024.
>
> [4] Dlf: Disentangled-language-focused multimodal sentiment analysis. AAAI 2025.
>
> **[Q2] Why was JS() chosen over other metrics (e.g., cosine or L2)?**
>
> [A2] We thank the reviewer for the question. Our token semantics are modeled as probability distributions, so a divergence operating in distribution space is generally more appropriate than cosine or L2, which are defined in Euclidean space and cannot capture uncertainty or distributional disagreement. While JS is not the only possible choice, it offers practical advantages—symmetry, boundedness, and stability near zero probabilities—making it better aligned with our setting than cosine or L2.
>
> A controlled comparison is conducted as follows. This indicates that, among the tested options, JS is the most effective for measuring token-level inconsistency in our framework.
>
> **Comparison of alternatives of JS(·) on the MOSI dataset in the setting of T+A.**
> | Method        | Acc-7 | Acc-2 | F1    | MAE   |
> |---------------|--------|--------|--------|--------|
> | **JS-div (Ours)** | **51.46** | **90.09** | **90.08** | **0.557** |
> | KL-div        | 50.32 | 89.30 | 89.20 | 0.612 |
> | Cosine        | 47.96 | 88.90 | 88.71 | 0.628 |
> | L2            | 47.23 | 88.42 | 88.06 | 0.698 |
>
> **[Q3] More clarification is needed in Equation (4).**
>
> [A3] Thanks for the detailed comment. We have revised Equation (4) to present the inter-modality consistency computation in a general pairwise form.
>
> $$\text{InterCS}(h_i) = \frac{1}{N} \sum_{n=1}^{N} \frac{1}{T} \sum^T_{j=1} \text{JS}(p_i^{\text{src}}|| p_{j}^{\text{align}}),$$ where:
>
> - $p_i^{\text{src}}$ and $p_{j}^{\text{align}}$ denote the sentiment prediction distributions of the $i$-th source token and the $j$-th aligned token, respectively.
>
> - $\text{JS}(\cdot || \cdot)$ is the Jensen-Shannon divergence, which measures the semantic discrepancy between the two distributions.
>
> - $T$ is the token length.
>
> - $N$ represents the number of modality pairs involved. For a setting with $M$ modalities, $N = M - 1$ (e.g., for 3 modalities, $N=2$).
>
> By minimizing this score, we enforce strict semantic alignment for each modality pair in the prediction space.
>
>
>
> **[Q4] Why does a 0.8 retention rate perform better?**
>
> [A4] Thanks for pointing out this important observation. The 1.0 setting passes all tokens after soft gating, whereas the 0.8 setting removes low-quality tokens entirely. This hard removal eliminates residual noise that soft gates cannot fully suppress, explaining why 0.8 slightly outperforms 1.0. In summary, consistency learning provides a soft, continuous regularization that identifies low-quality tokens, while hard removal eliminates them entirely. The small performance gain at a 0.8 retention ratio, therefore, reflects the complementarity of soft consistency learning and discrete hard pruning. This reinforces, rather than contradicts, the effectiveness of our consistency learning method.
>
> In conclusion, thanks again for your insightful and detailed comments. We hope our feedback will align with your expectations. It will be very appreciated if you would raise your rating.

---

> ### Author Response · Authors · 2025-11-17
> **Further Response to comments from Reviewer UzNZ**
>
> We sincerely appreciate your thoughtful feedback and the time devoted to our submission.
>
> We would like to concisely clarify the core innovation of our work. While redundancy and cross-modal conflicts have long been recognized as central challenges in multimodal sentiment analysis, existing methods address them only implicitly—through fusion heuristics, or decoupled architectural design—but none provides an explicit modeling mechanism for semantic consistency itself.
>
> Our work introduces a new perspective: we formulate consistency directly at the semantic distribution level, defining both intra- and inter-modality consistency through the proposed I²CS score, a token-level, distribution-based measure that is mathematically grounded and operational in practice. This explicit formulation allows the model to identify redundant or conflicting tokens in a principled way and to leverage consistency as a unified signal for both improved accuracy and efficient token selection. To the best of our knowledge, this is the first explicit, generalizable, and distribution-level consistency modeling framework in MSA.
>
> If the clarifications and additional analyses in the rebuttal satisfactorily address the raised concerns, we would be grateful if you could increase the rating. We are happy to provide any further clarification if needed.

---

> ### Author Response · Authors · 2025-11-28
>
> Dear reviewer UzNZ,
>
> We hope this message finds you well. We have carefully responded to all of the comments and clarifications requested in the reviews. If there are any remaining concerns, we would greatly appreciate it if you could kindly let us know so that we can further clarify. If our responses have addressed your questions, we would be very grateful if you could consider updating your scores accordingly.
>
> Thank you again for your time and feedback.

---

### Official Review · Reviewer_iZXh · 2025-11-01

**Soundness:** 3
**Presentation:** 4
**Contribution:** 3
**Rating:** 6
**Confidence:** 5

**Summary:**

The paper presents an I2C, a framework that explicitly models Intra- and Inter-modality Consistency to guide effective and efficient sentiment prediction. It first projects token-level features into a shared sentiment space and computes intra- and inter-modality consistency scores (I2CS).   I2C maintains comparable performance, exhibiting remarkable robustness across varying token budgets.

**Strengths:**

The paper is well-structured and logically compelling. It provides a clear explanation of the algorithms and offers a comprehensive set of experiments that thoroughly validate the proposed method.

**Weaknesses:**

It should be noted that the I2C method, which models Intra- and Inter-modality Consistency for feature representation, has been previously released in previous papers. Therefore, this work cannot claim to be its first proposer, which significantly limits its novelty. The Intra- and Inter-modality Consistency approach itself is more of an engineering heuristic and lacks substantial theoretical underpinnings. While it succeeds in improving experimental metrics, its conceptual novelty is relatively weak.

**Questions:**

1. The paper presents a framework that explicitly models Intra- and Inter-modality Consistency to guide effective and efficient sentiment prediction. The idea is very similar to the following framework, which effectively captures discriminative intra-frame and inter-frame features for representative feature learning.
It is suggested to refer to and analyze their similarity and differences.
Relation-mining self-attention network for skeleton-based human action recognition, Pattern Recognition, Vol. 139, 109455, 2023.
2. I2CS(hi) is calculated between every two modalities. The question is, has every pair of modalities, text, visual, and audio, being paired and calculated the I2CS(hi) value? Equations (4) and (5) do not show the detailed information.
3. About the performance of I2CS(hi), it is better to evaluate the contribution from every two-modality pair, and also illustrate the relationship between different modalities.

---

> ### Author Response · Authors · 2025-11-14
> **Response to comments from Reviewer iZXh**
>
> We greatly appreciate the recognition of our paper’s structure, clarity of algorithmic presentation, and the thoroughness of our experimental validation. Your comments are highly encouraging and help affirm the strengths of our work. Below, we address each of your questions and concerns in detail.
>
> **[Q1] Comparison with RSA-Net (Pattern Recognition, 2023)?**
>
> [A1] Thanks for pointing out RSA-Net. Regarding RSA-Net, we acknowledge its contribution to utilizing attention mechanisms for spatial-temporal relation mining in action recognition. However, a key distinction lies in the operational level: standard attention mechanisms (like in RSA-Net) typically model relationships in the intermediate feature space.
> In contrast, our I²C targets the specific challenges of MSA (e.g., modal redundancy and sentiment conflicts) by enforcing consistency in the prediction distribution space (high-level semantic space). This allows our model to align distinct modalities based on their sentiment semantics rather than just their low-level features. We have revised the Related Work section to explicitly highlight this fundamental difference between RSA-Net and our method.
>
> Furthermore, empirical results support this theoretical analysis. As shown in Table 3(b), we compared our method against an attention-based masking strategy, and the results demonstrate that our approach consistently outperforms the attention-based baseline, further validating the effectiveness of modeling consistency in the semantic space.
>
>
> **[Q2] Pairwise Modality Interaction (I2CS(hi)).**
>
> [A2] Yes, our approach models pairwise interaction by evaluating the distributional consistency between a source token and its aligned counterparts in other modalities. Mathematically, the pairwise interaction is quantified as follows: $$\text{InterCS}(h_i) = \frac{1}{N} \sum_{n=1}^{N} \frac{1}{T} \sum^T_{j=1} \text{JS}(p_i^{\text{src}}|| p_{j}^{\text{align}}),$$ where:
>
> - $p_i^{\text{src}}$ and $p_{j}^{\text{align}}$ denote the sentiment prediction distributions of the $i$-th source token and the $j$-th aligned token, respectively.
>
> - $\text{JS}(\cdot || \cdot)$ is the Jensen-Shannon divergence, which measures the semantic discrepancy between the two distributions.
>
> - $T$ is the token length.
>
> - $N$ represents the number of modality pairs involved. For a setting with $M$ modalities, $N = M - 1$ (e.g., for 3 modalities, $N=2$).
>
> This formulation ensures that the interaction is computed in a pairwise manner: for each token in the source modality, the model iteratively computes its consistency with the aligned tokens in every other modality. By minimizing this score, we enforce strict semantic alignment for each modality pair in the prediction space.
>
> In the revision, we have slightly modified Equation (4) and given a detailed description to make it concrete (N =1 for T+A or N=2 for T+A+V). Thanks again for the reminder.
>
> **[Q3] Contribution of Each Modality Pair？**
>
> [A3] We appreciate this suggestion. We compared the contribution of each modality pair as shown in the following table. It is easily observed that text–audio consistency contributes most to overall improvement, while audio–visual is complementary.
>
> **Ablation of I²CS pair on the MOSI dataset in the setting of T+A+V.**
>
> | Modality pair | Acc-7 | Acc-2 | F1    | MAE   |
> |-----------|-------|-------|-------|-------|
> | (A, V)    | 43.44 | 83.58 | 83.02 | 0.771 |
> | (V, T)    | 45.92 | 84.49 | 84.50 | 0.732 |
> | (T, A)    | 47.23 | 85.26 | 85.21 | 0.728 |
> | **All (Ours)** | **47.96** | **86.06** | **86.06** | **0.698** |
>
>
> In conclusion, thanks again for your insightful and detailed comments. We hope our feedback will align with your expectations. It will be very appreciated if you would raise your rating.

---

> ### Author Response · Authors · 2025-11-17
> **Further Response to comments from Reviewer iZXh**
>
> We sincerely appreciate your thoughtful feedback and the time devoted to our submission.
>
> We would like to concisely clarify the core innovation of our work. While redundancy and cross-modal conflicts have long been recognized as central challenges in multimodal sentiment analysis, existing methods address them only implicitly—through fusion heuristics, or decoupled architectural design—but none provides an explicit modeling mechanism for semantic consistency itself.
>
> Our work introduces a new perspective: we formulate consistency directly at the semantic distribution level, defining both intra- and inter-modality consistency through the proposed I²CS score, a token-level, distribution-based measure that is mathematically grounded and operational in practice. This explicit formulation allows the model to identify redundant or conflicting tokens in a principled way and to leverage consistency as a unified signal for both improved accuracy and efficient token selection. To the best of our knowledge, this is the first explicit, generalizable, and distribution-level consistency modeling framework in MSA.
>
> If the clarifications and additional analyses in the rebuttal satisfactorily address the raised concerns, we would be grateful if you could increase the rating. We are happy to provide any further clarification if needed.

---

> ### Author Response · Authors · 2025-11-28
>
> Dear reviewer iZXh,
>
> We hope you are doing well. We have carefully responded to all of the comments and clarifications requested in the reviews. If there are any remaining concerns, we would greatly appreciate it if you could kindly let us know so that we can further clarify. If our responses have addressed your questions, we would be very grateful if you could consider updating your scores accordingly.
>
> Thank you again for your time and feedback.

---

### Author Response · Authors · 2025-11-27
**Wait for the feedback from Reviewers**

Dear reviewers,

We hope you are doing well.

We sincerely appreciate your time devoted to our submission. We believe we have addressed the questions raised in the reviews in our response. If there are any remaining concerns, we would greatly appreciate it if you could kindly point them out in detail so we can further clarify; if our responses have resolved your concerns, we would be grateful if you could consider updating your scores accordingly. Your feedback is very critical to this work.

Thanks a lot!

Authors

---

### Meta-Review · Area_Chair_rzdx · 2026-01-09

**Summary:**

The paper proposes I²C, a framework for Multimodal Sentiment Analysis that models intra- and inter-modality consistency in the sentiment prediction distribution space to mitigate redundancy and conflicts. While the experimental performance is competitive, the decision is rejection. The primary reasons include concerns regarding limited conceptual novelty (viewed as an engineering heuristic rather than a theoretical breakthrough), insufficient justification for key design choices (e.g., masking after self-attention), and varying levels of clarity in the presentation and mathematical definitions.

**Reviewer Concerns:**

Addressed: The authors provided necessary clarifications regarding the formulation of the Inter-modality Consistency Score (Equation 4), confirmed the non-usage of CLS tokens to ensure modality comparability, and provided additional ablations justifying the choice of Jensen-Shannon divergence over other metrics.

Outstanding: Significant concerns remain regarding the novelty and theoretical depth. Reviewer iZXh noted the approach resembles existing attention mechanisms (like RSA-Net) and lacks substantial theoretical underpinnings, appearing as an "engineering heuristic." Furthermore, Reviewers geLK and UzNZ maintained doubts about the fundamental logic of the architecture (e.g., the effectiveness of masking after encoder interaction) and why the proposed soft consistency learning is insufficient without hard pruning (0.8 ratio > 1.0), suggesting the motivation needs stronger theoretical support.

**Reviewer Scores:**

Reviewer iZXh (6): Likely would have maintained the score. While specific questions were answered, the criticism regarding the method being an "engineering heuristic" limits the potential for a higher score.

Reviewer UzNZ (4): Likely would have remained at 4 or marginally 5. The rebuttal regarding the retention ratio (where removing data improves performance) inadvertently highlighted potential inefficiencies in the core soft-gating mechanism, which may not have fully alleviated concerns about robustness.

Reviewer geLK (2): Likely would have remained low (2 or 3). The reviewer's impression that the paper is "unfinished" and the fundamental disagreement with the architectural logic (post-encoding masking) are difficult to resolve solely through a rebuttal text without a major revision of the manuscript.

---

### Decision · Program_Chairs · 2026-01-26

Reject